# Analysis of Mechanical Properties and Mechanism of Natural Rubber Waterstop after Aging in Low-Temperature Environment

**DOI:** 10.3390/polym13132119

**Published:** 2021-06-28

**Authors:** Lin Yu, Shiman Liu, Weiwei Yang, Mengying Liu

**Affiliations:** 1College of Mechanics and Materials, Hohai University, Nanjing 210098, China; ntyangweiwei@163.com (W.Y.); liumengying71@163.com (M.L.); 2Jianke Engineering Consulting Co., Ltd., Shanghai 200032, China; liushiman0@163.com

**Keywords:** rubber waterstop, low temperature, aging performance, microscopic examination, mechanism analysis

## Abstract

In order to elucidate the aging performance and aging mechanism of a rubber waterstop in low-temperature environments, the rubber waterstops were placed in the freezing test chamber to accelerate aging, and then we tested its tensile strength, elongation, tear strength, compression permanent deformation and hardness at different times. Additionally, the damaged specimens were tested by scanning electron microscope, Fourier transform infrared spectroscopy and energy dispersive spectrometry. The results showed that with the growth of aging time, the mechanical properties of the rubber waterstop are reduced. At the same time, many protrusions appeared on the surface of the rubber waterstop, the C element gradually decreased, and the O element gradually increased. During the period of 72–90 days, the content of the C element in the low-temperature air environment significantly decreased compared with that in low-temperature water, while the content of O element increased significantly.

## 1. Introduction

In water conservancy and hydro-power projects, considering that concrete cannot be continuously poured, construction joints, settlement joints and deformation joints are needed in order to adapt to the deformation of foundation and the deformation caused by the change in temperature. The rubber waterstop, as the most commonly used waterstop material, can effectively prevent the leakage and seepage of building joints, and play the role of shock absorption and buffer, so it is widely used in projects [1,2,3,4]. In applications, rubber waterstop is often exposed to various environments, such as oxygen, ozone, light and temperature, so it often changes in composition and structure [5,6].

At the same time, natural rubber has a huge advantage over fossil-based polymers in terms of environmental friendliness. First of all, natural rubber is derived from rubber trees, while fossil-based polymers are derived from petroleum by-products, which means that natural rubber is inexhaustible and fossil-based polymers are limited. Meanwhile, fossil-based polymers easily burn and release gases that are harmful to the environment, such as toluene. Secondly, the performance of fossil-based polymers in degradability and recycling is worse than that of natural rubber. As fossil-based polymers are difficult to degrade [7], many fossil-based polymers products are discarded and flow into the sea [8]. This causes serious pollution to the marine environment and kills a large number of marine organisms, but natural rubber products can be recycled, and many countries have established the corresponding regulations [9].

Deng Jun et al. [10] studied the effects of different aging temperatures and aging times on the properties of different types of rubber. The results showed that with the increase in aging temperature and aging time, the rubber hardness increased and its tensile strength decreased. Li Bo et al. [11] accelerated the aging of rubber in a hot oxygen environment and investigated the changes in the mechanical properties of rubber before and after aging and the degree of aging with aging temperature and time. The results showed that the elongation and fracture stress of rubber gradually decreased with increasing aging time and aging temperature, and a small increase in temperature would lead to a significant decrease in mechanical properties. J.R. Beatty et al. [12] studied the influence of time and pressure on rubber in a low-temperature environment. The results showed that the hardness of rubber material at low-temperature increased with the gradual increase in time and pressure and proposed that one of the reasons for the increase in rubber hardness was crystallization.

In the above studies, the environment of rubber aging was mostly set to high temperature and high oxygen or lack of microscopic testing and mechanism analysis. However, the temperature in Xigaze is low all year round, with an average temperature of 6.5 °C [13]. The average temperature in the coldest month (January) is −3.2 °C while the average temperature in the hottest month (July) is 14.6 °C [14]. Additionally, the rubber waterstop used in water conservancy projects is generally poured inside the concrete, which greatly reduces the contact with oxygen and ozone, and also shelters from light [15]. Therefore, we put the rubber waterstop in the freezing test chamber for accelerated aging, and tested its tensile strength, elongation, tear strength, compression set and hardness under different aging times, and carried out microscopic detection on the damaged specimen, trying to find the aging mechanism of rubber in the freezing environment.

## 2. Experiment Flow Chart

Figure 1 shows the flow chart of our experiments.

## 3. Experimental Program

### Materials

The BW5 rubber waterstop with the specification of 300 mm × 8 mm used in the construction joint of the real project was employed in this study. The vulcanized rubber used in this study mainly consisted of polyisoprene (92–95% of cis1,4-polyisoprene) which was produced by Hebei Jingjia Rubber products Co., Ltd. (China). The main components of the rubber waterstop are natural rubber, insoluble sulfur(S), accelerator, zinc oxide (ZnO), stearic acid (SA) and carbon black.

## 4. Sample Preparation and Aging Environment

Five groups of specimens were prepared in two environments, and each group consisted of three tensile specimens, five tear specimens, three permanent compression deformation specimens and one hardness specimen. Tensile specimens were I type dumbbell shaped with a thickness of 2.0 mm ± 0.2 mm as shown in Figure 2. The tear specimen was of right-angle type with a thickness of 2.0 mm ± 0.2 mm as shown in Figure 3. The compression permanent deformation specimen was a B-type cylinder with a diameter of 13.0 mm ± 0.5 mm.

Considering the constant change of the dam water level in this project [14], the rubber waterstop buried in the dam may be above or below the water surface in different seasons, so the low-temperature environment is divided into low-temperature air environment and low-temperature water environment. Since the lowest annual temperature in Xigaze is −25 °C [16], the low-temperature environment in this paper was set at −25 °C.

## 5. Equipment

The DW40 refrigerating test chamber was produced by Nanjing Sibenke Experiment Co (China). The KSL-10KN electronic universal testing machine was produced by YangZhou KaiDe Experiment Co (China). The YF-8426 compression permanent deformation machine was produced by YangZhou YuanFeng Experiment Co (China). The Quanta 250F SEM was produced by FEI (Hillsboro, OR, USA). The Nexus 470 Fourier infrared spectrometer was produced by Nicolet (Waltham, MA, USA).

## 6. Testing Methods

### 6.1. Mechanical Properties Test

The temperature of the freezer test chamber was set to −25 °C, and then the 5 groups of specimens were aged in the freezing test chamber for 18 days, 36 days, 54 days, 72 days and 90 days, respectively. After reaching the aging time, the specimens were taken out and placed indoors for 24 h. The tensile strength, elongation and tear strength were tested by an electronic universal tensile testing machine. The loading speed of the testing machine was set to 500 mm/min [17,18]. The shore hardness tester was used to test the hardness of the specimen [19]. The compression specimen was put into the compression permanent deformer to make its compression rate reach 23–28%. After 168 h of compression, the specimen was immediately released and allowed to recover at room temperature for 27–33 min, and then measured the height of the specimen [20]. The compression permanent deformation was calculated according to Equation (1):(1)$$C=\frac{h_{0}-h_{1}}{h_{0}-h_{s}}\times 100\%$$
where *C*—compression set (%); *h*_0_—initial height (mm); *h*_1_—height after recovery (mm); and *h_s_*—limiter height (mm).

### 6.2. Scanning Electron Microscopy with X-Ray Microanalysis (SEM-EDS) Test

The microstructure and elemental analysis of the rubber waterstop after accelerated aging treatment were investigated using SEM (FEI quanta 250F) equipped with an EDS at an accelerating voltage of 30 kV. The test pieces were then observed at 500, 1000 and 6000 magnifications, and the C and O element contents were analyzed by EDS at a magnification of 1000.

### 6.3. Fourier Infrared Spectroscopy-Attenuated Total Reflection Total Reflection (FTIR-ATR) Test

The FTIR spectra were recorded using a Nicolet Nexus 470 spectrometer equipped with an ATR attachment. The FTIR was operated within the scan angle range of 5°–90°and a scan speed of 5°/min.

## 7. Results and Discussion

### Mechanical Properties

After aging for different times in the low-temperature air environment and low-temperature water environment, the mechanical properties of the rubber waterstop are shown in Table 1 and Table 2.

It can be seen from Table 1 and Table 2, Figure 4 and Figure 5 that the tensile strength, elongation, tear strength and compression permanent deformation of the rubber waterstop decreased in the low-temperature environment. Tensile strength and tear strength decreased by 34.9% and 24.7% in the low-temperature air environment. Tensile strength and tear strength decreased by 19.1% and 23.3% in the low-temperature water environment. This shows that the brittleness of rubber increases with the aging time, and the adverse effect is more obvious in the low-temperature air environment. The hardness increases slowly, the material becomes hard and gradually loses its elasticity. The decrease in compression permanent deformation indicates that the deformation capacity of vulcanized rubber gradually decreases with the increase in aging time at low temperature, that is, the plasticity weakens. The reason for this is that the rubber main chain and side chain of the molecular chain and crosslinking chain fracture happens, simultaneously creating new crosslinking, whereas in the low-temperature environment, the rubber molecular chain is in a new crosslinking reaction, thus showing surface hardening after aging and brittle crack, namely due to its decreasing tensile strength, elongation, tear strength and compression permanent deformation.

## 8. Failure Modes

### Surface Damage

After aging at low temperature, the surface of the specimen significantly changed. The surface of the specimens without aging was relatively flat, and the micro-pits visible to the naked eye appeared on the surface of the specimens after 36 days of low-temperature treatment, and the number increased with time. At 90 days, pits and tiny holes appeared on the surface of the specimen in the water. At low temperature, the surface of the specimen became rough while the specimen became hard. The apparent morphology of the rubber is shown in Figure 6.

The surface morphology of the rubber waterstop before and after aging is further examined by SEM and is shown in Figure 7.

As shown in Figure 7a, the virgin surface of the rubber waterstop was flat, with white and black additive particles uniformly distributed. As shown in Figure 7b, after 36 days aging under low-temperature air environment, the additive particles on the surface completely disappeared, but many protrusions appeared, and the surface of the specimen became abnormally uneven. With the increase in aging time, the protrusions increased. After 90 days, tiny holes appeared on the surface of the specimen, as shown in Figure 7c. As shown in Figure 7d–f, in the low-temperature water environment, the surface morphology of the specimens protruded, and after 54 days, the rubber surface appeared stepped stratification. After 90 days, the surface became rougher as the protrusion on the surface increased, and the stepped stratification phenomenon became more obvious.

## 9. Cross-Sectional Damage of Specimens after Tensile Test

Figure 8 shows the cross-sectional morphologies of the tensile test pieces in the low-temperature air environment and low-temperature water environment. The cross-sectional morphologies after 18 days and 90 days in low-temperature air environment are shown in Figure 8b,c, as there were more step-by-step undulations on the surface, and no additive particles. The cross-sectional morphologies changed little in the aging cycle, indicating that the environment has little effect on the adhesion of the rubber matrix. The cross-sectional morphologies after 18 days, 54 days and 90 days in the low-temperature water environment are shown in Figure 8d–f, where a small number of additive particles appeared after 90 days, indicating that this environment can reduce the adhesion of the rubber matrix. Thus, this environment has an effect on the strength of the rubber waterstop, but the effect is far less than that in the low-temperature air environment.

## 10. Chemical Analysis

### 10.1. Energy-Dispersive Spectrometer Analysis

The changes in the content of C and O elements (wt %) in the rubber waterstop samples are shown in Table 3.

It can be seen that increasing the aging time led to a decrease in the C element and a concurrent increase in the O element in the rubber waterstop. In the low-temperature air environment, the content of the C element decreased by 4.77% while the content of O element increased by 45.0% after 72 days; the oxidation reaction suddenly became faster from 72 days to 90 days, the content of C element decreased by 13.98% and the content of the O element increased by 105.0% after 90 days. In the low-temperature water environment, the content of C element decreased by 4.16% while the content of the O element increased by 61.19% after 90 days. Intramolecular reactions can occur in vulcanized rubber polymers due to the polyunsaturated property and the short distance between the double bonds. In the aging process of these polymers, the C–C double bond breaks and forms hyperoxide-hydroperoxides, which leads to the decrease in the C element and the increase in the O element on the rubber surface.

Figure 9 shows the change curve of the C and O element content with aging time under three different environments. The x axis represents the aging time, and the y axis represents the content of the elements. Compared with the natural environment, the oxidation reaction of the rubber waterstop in the low-temperature environment was slower. This indicates that low temperature is helpful to alleviate the aging of rubber and the weakening of the adhesion of the rubber matrix.

### 10.2. Fourier Infrared Spectroscopy-Attenuated Total Refection Analysis

The infrared spectra of the rubber waterstop in low-temperature air environment and low-temperature water environment after 18 days, 36 days, 54 days, 72 days and 90 days are shown in Figure 10 and Figure 11.

Figure 10 presents the infrared spectra of the rubber waterstop under a low-temperature air environment. In the ATR-FTIR spectra, the peak at 2900–2800 cm^−1^, 1430 cm^−1^ and 1372 cm^−1^ was assigned to C–H stretching vibrations of –CH_2_–, –CH_3_–. The peaks at 1640 cm^−1^ and 878 cm^−1^ were defined as =C–H stretching vibration of the cis-1,4 structure, while the peak at 1443 cm^−1^ was assigned to deformation vibration absorption of –C–O–. It shows that the C–H bond and =CH slowly weaken, and –CO– appears on the molecular chain. The most obvious change was the stretching vibration peak of hydroxyl (O–H) at 3500–3200 cm^−1^. After 54 days, the peak intensity sharply increased, and then slowly increased. According to the changes of the above spectral peaks, it can be concluded that the double bond of the rubber waterstop slowly decreases and the product of the vicinal diol obviously increases after aging in the low-temperature air environment, which is consistent with the aging law in the natural environment.

In the low-temperature water environment, the infrared spectrum of rubber after aging is similar to that in the low-temperature air environment. With the increase in aging time, the peak intensities of –CH_3_– at 2915 cm^−1^ and 2841 cm^−1^ and C=C at 1640 cm^−1^ and 878 cm^−1^ decreased slightly, while the peak intensities of –CH_2_– at 1430 cm^−1^ and CH_3_ at 1372 cm^−1^ increased slightly. The peak value of –C–O– group at 1002 cm^−1^ obviously decreased, and the change in peak value at 3237 cm^−1^ was the same as that in the dry environment.

From the analysis of Figure 10 and Figure 11, it can be found that during the low-temperature aging test, the basic structure of the rubber and the position of the characteristic peak do not significantly change before and after aging, which indicates that during the low-temperature treatment process of 0–90 days, the basic chemical structure of the rubber waterstop was not obviously damaged or the damage is relatively small.

## 11. Conclusions

Based on the findings from this study, the following conclusions can be drawn:Increase in low-temperature aging time leads to a decrease in the tensile strength, elongation, tear strength and compression permanent deformation and an increase in the hardness of the rubber waterstop.SEM observation reveals that with the increase in aging time, the surface additive particles disappear, many protrusions appear and gradually increase in the low-temperature air environment. In the low-temperature water environment, the surface protrusions become rough, and there are only a small number of additive particles in the cross-section at 90 days, which reduces the adhesion of the rubber matrix and affects the strength of the rubber waterstop.FTIR-ATR analysis shows that the infrared spectra of the low-temperature air environment and low-temperature water environment are almost the same, the decrease in functional groups used to characterize the basic structure of polyisoprene is not obvious during the aging process, and the peak amplitude produced by the hydroxyl carboxyl group and other functional groups during the oxidation process is small, and the oxidation is slow.EDS results show that with the increase in aging time, there is an increase in O element and a decrease in C element in the low-temperature air environment. In the low-temperature water environment, the rubber matrix still maintains good adhesion.

## Figures and Tables

**Figure 1 polymers-13-02119-f001:**
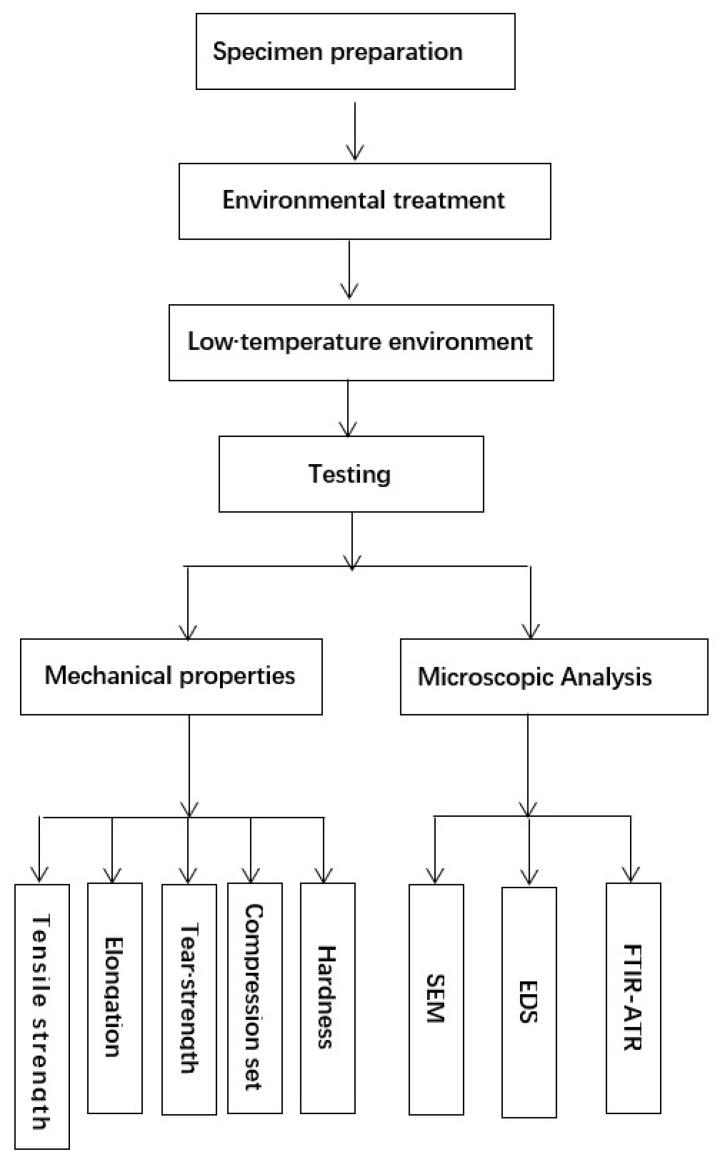
Experiment flow chart.

**Figure 2 polymers-13-02119-f002:**
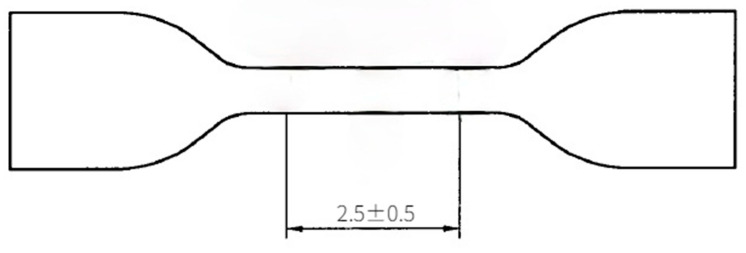
Dumbbell type specimen.

**Figure 3 polymers-13-02119-f003:**
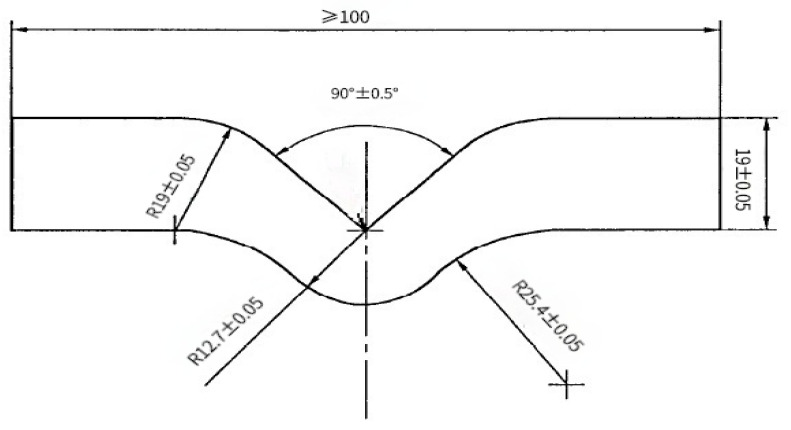
Right-angle specimen.

**Figure 4 polymers-13-02119-f004:**
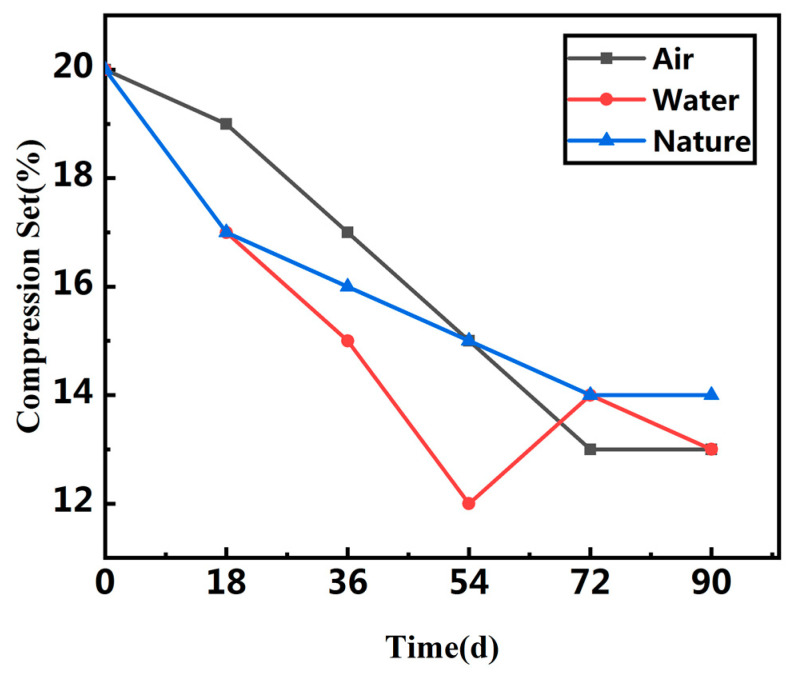
Compression set of waterstop in three different environments.

**Figure 5 polymers-13-02119-f005:**
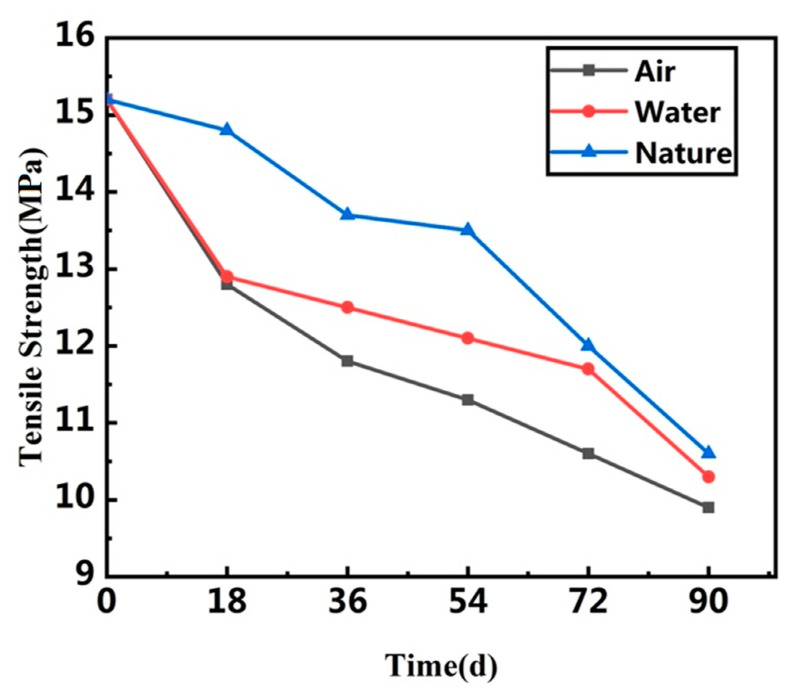
Tensile strength of waterstop in three different environments.

**Figure 6 polymers-13-02119-f006:**
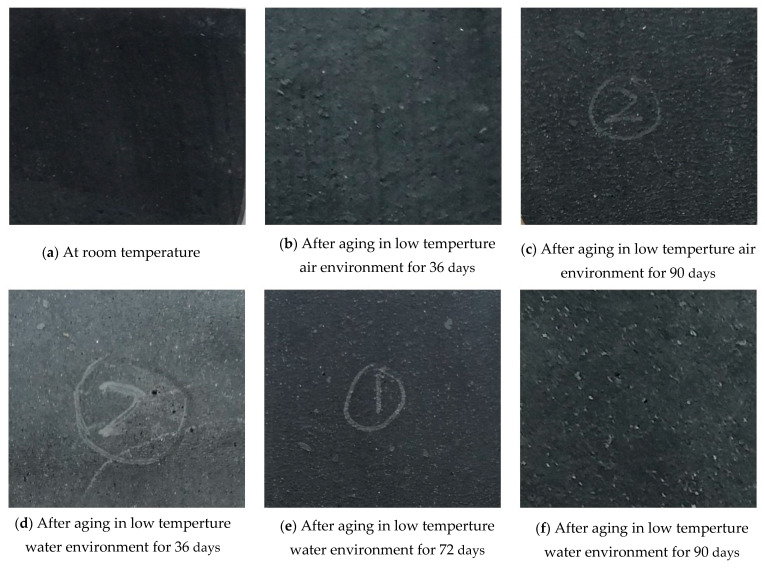
Apparent morphology of the specimen after aging at low temperature.

**Figure 7 polymers-13-02119-f007:**
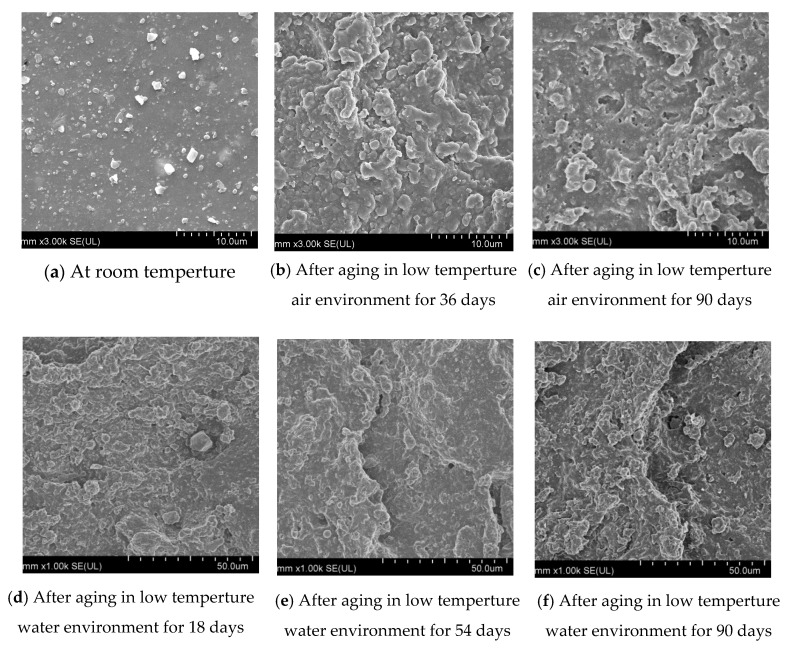
Surface micromorphology.

**Figure 8 polymers-13-02119-f008:**
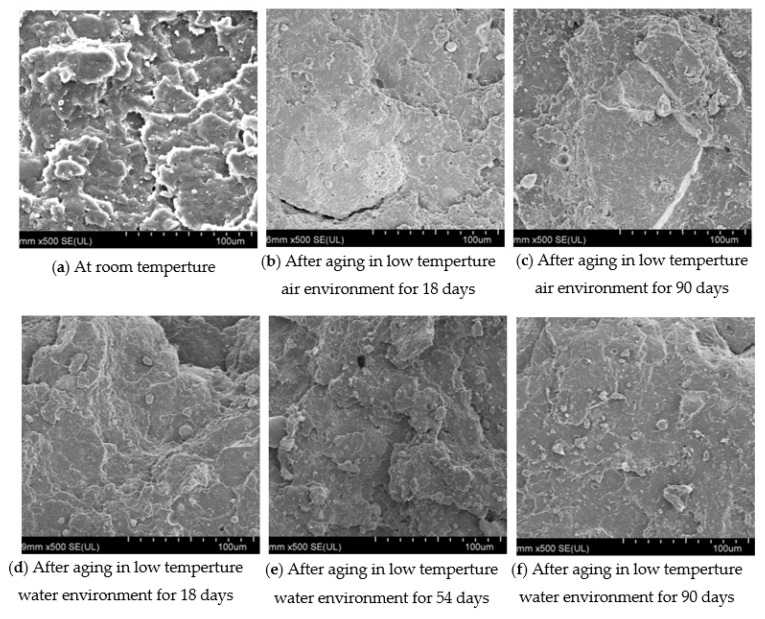
Micromorphology of section.

**Figure 9 polymers-13-02119-f009:**
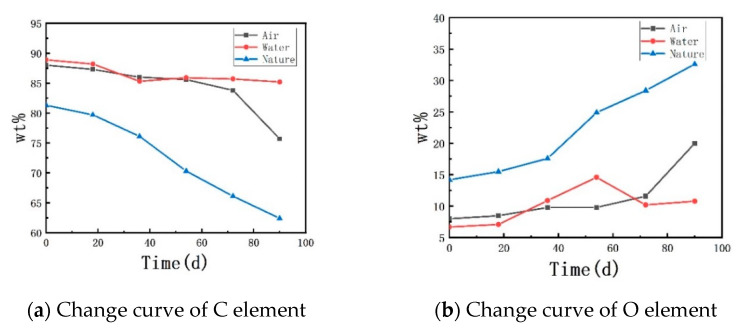
Change curve of element content.

**Figure 10 polymers-13-02119-f010:**
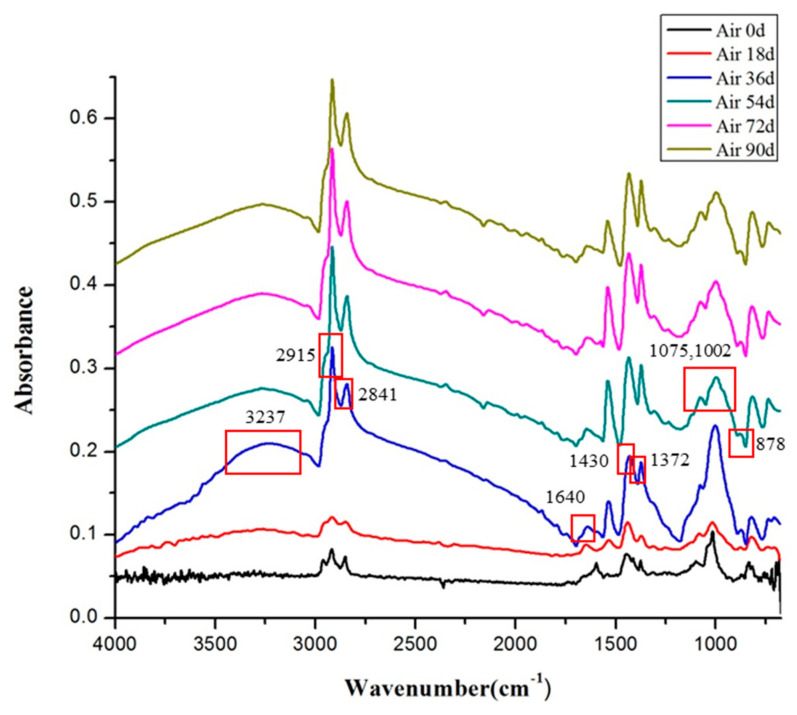
Infrared spectra of low-temperature air environment.

**Figure 11 polymers-13-02119-f011:**
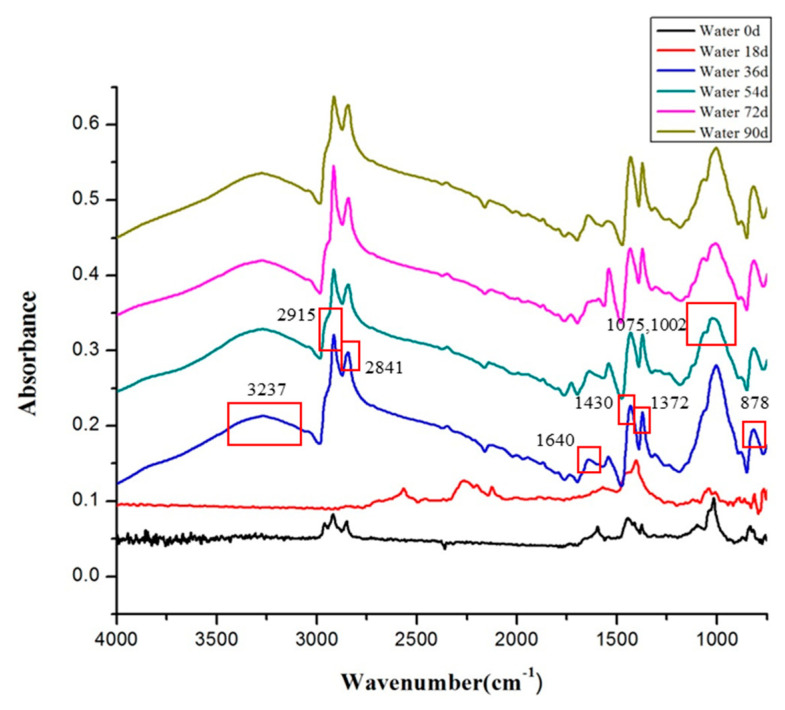
Infrared spectra of low-temperature water environment.

**Table 1 polymers-13-02119-t001:** Mechanical properties of aging rubber waterstop in low-temperature air environment.

Time(d)	Tensile Strength (MPa)	Elongation (%)	Tear Strength (kN/m)	Hardness (HA)	Compression Set (%)
0	15.2	494.9	55.9	56	20
18	12.8	482.1	52.2	60	19
36	11.8	391.8	51.0	60	17
54	11.3	342.0	48.6	60	15
72	10.6	339.4	46.9	61	13
90	9.9	335.2	42.1	61	13

**Table 2 polymers-13-02119-t002:** Mechanical properties of aging rubber waterstop in low-temperature water environment.

Time(d)	Tensile Strength (MPa)	Elongation (%)	Tear Strength (kN/m)	Hardness (HA)	Compression Set (%)
0	15.2	494.9	55.9	56	20
18	12.9	466.7	47.8	59	17
36	12.5	414.7	46.9	59	15
54	12.1	398.0	45.4	59	12
72	11.7	360.1	44.6	59	14
90	12.3	359.4	42.9	60	13

**Table 3 polymers-13-02119-t003:** Changes in C and O element content of different samples.

	Time (d)	0	18	36	54	72	90
Element							
C_(Air)_		88.0	87.3	86.0	85.6	83.8	75.7
O_(Air)_		8.0	8.5	9.8	9.8	11.6	20.0
C_(Water)_		88.9	88.2	85.3	85.9	85.7	85.2
O_(Water)_		6.7	7.1	10.9	14.6	10.2	10.8

## Data Availability

The data presented in this study are available on request from the corresponding author.

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
