# Peer review of "Analysis of Mechanical Properties and Mechanism of Natural Rubber Waterstop after Aging in Low-Temperature Environment"

_polymers, 2021, doi:10.3390/polym13132119_

Round 1
Reviewer 1 Report
- Summary, strengths, weaknesses, overall contribution
In the paper the Authors study the changing trend of various mechanical properties of rubber waterstop with time in low-temperature environment and carry on the mechanism analysis. Unfortunately, the objective is not precisely stated and overall contribution is unclearly communicated in the paper, however after major revision, it can be interesting for rubber community.
- Major comments
The paper may be reconsidered for publication if the authors will refer to the following remarks and do the necessary corrections, which would significantly improve the paper:
B1. In Introduction, there is a lack of paragraph where the paper itself is described as well as the aim of the research is not clearly and directly given. The Authors should also discuss the role of adhesion of the additive particles on the mechanical properties of the rubber. The Authors may refer to the following papers: DOI: 10.1007/s00542-013-2036-0; 10.1016/j.ceramint.2019.08.063; 10.1007/s11249-020-01296-8
B2. Figure 1 and 2 – the dimensions are totally unreadable.
B3. It is not clear what is exactly shown in Fig. 3?
B4. In Fig. 4 the scale is invisible. What is marked with a red circle?
B5. In Fig. 5 the scale is invisible.
B6. In Fig. 6 the axis descriptions are invisible. The Authors should indicate the measurement errors.
B7. In my opinion the tensile and compression curves should be shown in the paper.
Author Response
Dear reviewer:
I am very grateful to your comments for the manuscript. According with your advice, we amended the relevant part in manuscript. Some of your questions were answered below.
1)In the introduction, I added the description of the paper and pointed out the purpose of the paper.
2)I redrawn figures 1 and 2 to make the dimensions clear。
3)In Fig.3, it exactly showed the apparent morphology of specimen after aging at low temperature air and water environments.
4、5) The scale of the figure is indicated in the lower right corner of the figure.
6) We have added the description of the axis.
7) The tensile and compression curves have be shown in the paper as the Fig.3 and Fig.4.
we would like to express our great appreciation to you for comments on our paper.Looking forward to hearing from you .
Thanks you and best regards.
Yours sincerely,
Corresponding writer: L.Yu
Email: [email protected]

Reviewer 2 Report
The paper presents an analysis of the mechanical properties and mechanism of rubber waterstop after aging in low-temperature environment
The following recommendations are proposed:
- Please, rewrite the abstract. The reviewer believes that this can be organized in a way to provide a wider audience with a better understanding of the work performed.
- I recommend introducing the research approach in a different section.
- Please, provide a flow chart of the research approach.
- Some parts of the figures are too small, blurry, and hard to understand. Also, fonts are not always large enough; for example, figure 2.
- Overall, English needs to be double-checked for typos.
- The paper is valuable from a practical point of view to engineers and practitioners and not only for research. Nice work!
Author Response
Dear reviewer:
I am very grateful to your comments for the manuscript. According with your advice, we amended the relevant part in manuscript. Some of your questions were answered below.
1) We rewrote the abstract and tried to give readers a better understanding of what we have done.
2)We provided a flow chart of the research approach.
3) We have modified some of the images so that they can be seen clearly.
4) We have reviewed and revised the spelling and usage of English.
we would like to express our great appreciation to you for comments on our paper.Looking forward to hearing from you .
Thanks you and best regards.
Yours sincerely,
Corresponding writer: L.Yu
Email: [email protected]

Round 2
Reviewer 1 Report
The Authors have addressed ale the issues rised in the review.
The paper should be accepted for publication.
Reviewer 2 Report
- Comments addressed